# Retinoic Acid Action in Cumulus Cells: Implications for Oocyte Development and In Vitro Fertilization

**DOI:** 10.3390/ijms25031709

**Published:** 2024-01-30

**Authors:** Neil Sidell, Augustine Rajakumar

**Affiliations:** Department of Gynecology & Obstetrics, Emory University School of Medicine, Atlanta, GA 30322, USA; augustine.rajakumar@emory.edu

**Keywords:** retinoic acid, oocyte, granulosa cells, in vitro fertilization, connexin 43

## Abstract

In the field of human in vitro fertilization (IVF), selecting the best oocyte for freezing or embryo for transfer remains an important focus of clinical practice. Although several techniques are and have been used for this goal, results have generally not been favorable and/or are invasive such that damage to some embryos occurs, resulting in a reduced number of healthy births. Therefore, the search continues for non-invasive oocyte and embryo quality markers that signal the development of high-quality embryos. Multiple studies indicate the important positive effects of retinoic acid (RA) on oocyte maturation and function. We previously showed that a high follicular fluid (FF) RA concentration at the time of oocyte retrieval in IVF protocols was associated with oocytes, giving rise to the highest quality embryos, and that cumulus granulosa cells (CGCs) are the primary source of follicle RA synthesis. Data also demonstrated that connexin-43 (Cx43), the main connexin that forms gap junctions in CGCs, is regulated by RA and that RA induces a rapid increase in gap junction communication. Here, we hypothesize that CGC RA plays a causal role in oocyte competency through its action on Cx43 and, as such, may serve as a biomarker of oocyte competence. Multiple studies have demonstrated the requirement for Cx43 in CGCs for the normal progression of folliculogenesis, and that the increased expression of this connexin is linked to the improved developmental competence of the oocyte. The data have shown that RA can up-regulate gap junction intercellular communication (GJIC) in the cumulus–oocyte complex via a non-genomic mechanism that results in the dephosphorylation of Cx43 and enhanced GJIC. Recognizing the positive role played by gap junctions in CGCs in oocyte development and the regulation of Cx43 by RA, the findings have highlighted the possibility that CGC RA levels may serve as a non-invasive indicator for selecting high-quality oocytes for IVF procedures. In addition, the data suggest that the manipulation of Cx43 with retinoid compounds could provide new pharmacological approaches to improve IVF outcomes in cases of failed implantation, recurrent miscarriage, or in certain diseases that are characterized by reduced fecundity, such as endometriosis.

## 1. The Search for High Quality Oocytes

In the field of human in vitro fertilization (IVF), selecting the best oocyte for freezing or embryo for transfer remains an important focus of clinical practice because it improves outcomes. In the earliest days of IVF, enhanced selection was achieved by extending the time embryos were cultured from day 2, when embryos have about four cells, to day 3, when they have about eight cells, to the current practice of transferring blastocysts on day 5 or 6 of culture. This extended culture substantially increased the chance for successful implantation. However, with this approach the implantation rate still rarely exceeds 50%, even in the youngest patients. Therefore, other approaches able to improve implantation rates by finding the best embryos continue to be pursued. Of these more recent attempts, preimplantation genetic testing has gained a foothold in clinical practice [1,2]. In this process, a small number of trophectoderm cells are removed from blastocysts, which are then assessed for ploidy status. When initially investigated, biopsies were performed on day 3 (removing 1 or 2 cells of the 8) and evaluated by “fluorescence in situ hybridization”. The results were not favorable. In more recent years, the biopsy procedure has been delayed to day 5 or 6 of culture when embryos have more cells (so the biopsy can be larger and have less impact), and the testing platform is typically next-generation sequencing, which is both more accurate and comprehensive. In this iteration, randomized trials have shown a shorter time to pregnancy and fewer miscarriages [2]. However, this approach, now called “preimplantation genetic testing for aneuploidy” (PGT-A), is not without risk [3]. Risks include damage to some embryos, the misdiagnosis of the true ploidy status (testing errors), false positives and negatives (due to mosaicism, the ploidy status of the inner cell may not be the same as the cells biopsied), and loss due to an extra cryopreservation step. It is therefore likely that PGT-A reduces the number of successful pregnancies from a set of embryos that would occur had this invasive testing not been performed. Therefore, the search continues for non-invasive oocyte and embryo quality markers that signal the best embryo [4,5]. This search has taken various paths. Some have examined the follicular fluid of individual follicles for distinct hormone [6], RNA or protein profiles [7]. A study of cumulus cell transcriptome expression found no impact on live birth rates in human IVF [8]. Time-lapse imaging of embryo development in the laboratory has also been conducted [9]. It is not clear whether any of these tests add predictive power to the current method of selection based on embryo morphology. In considering other non-invasive approaches, we have focused on factors that support healthy oocyte growth in vivo, in the hope that this feature might not only predict improved outcomes, but also that it might be exploited therapeutically to directly improve oocyte potential. Recognizing that CGC gap junctions are composed predominantly of connexin 43 (Cx43) and play an important role in oocyte development [10,11], we are proposing that the production of RA in CGC via its regulation of Cx43 may serve as a marker of oocyte quality that will give rise to high-quality embryos and subsequent successful pregnancies.

## 2. Retinoic Acid Levels in Cumulus Cells as a Non-Invasive Predictor of Oocyte Quality

Numerous pieces of evidence highlight the significant impact of vitamin A compounds (retinoids) on oocyte maturation and function [12,13,14,15]. The in vivo administration of retinoids in animals including pigs, sheep, and cattle has demonstrated enhanced oocyte fertilization competence [13,16,17], while in vitro studies have identified all-trans retinoic acid (RA) as the active vitamin A metabolite in this process [18,19,20]. Recognizing that cumulus granulosa cells (CGCs) have tight connections to oocytes via their gap junctions, and based on prior work indicating a positive role for these connections in oocyte development [10,11], more recent studies have focused on the actions of RA to support this intercellular communication through its effect on Cx43 [21,22,23,24]. Concentrations of retinol (ROL, the substrate for RA synthesis) in the follicular fluid (FF) of women and animals have been reported by several laboratories [25,26,27], and our group directly quantified the FF levels of RA [28]. This work demonstrated that high follicular fluid RA concentrations at the time of oocyte retrieval in IVF protocols were associated with oocytes giving rise to the highest quality embryos [28]. Importantly, for follicles from which a mature egg was retrieved (“mature egg follicle”), an increase in the percentage of grade I embryos was observed across the tertiles of RA distribution; 57% of the embryos generated from mature oocytes in the highest RA tertile were classified as grade I versus only 18% of those derived from mature oocytes in the lowest tertile (Figure 1). As such, the follicular fluid RA concentration appeared to be an additional discriminating factor beyond the usual morphologic criteria that could be used for predicting successful oocyte fertilization and the generation of high-quality embryos. Subsequently, we quantified RA and other retinoids in primary CGCs from women undergoing IVF [21]. The findings indicate elevated retinoid levels in CGCs and the active synthesis of RA from its ROL substrate. These results were compared with the retinoid levels in human endometrium and mammary glands, tissues known for active RA synthesis from ROL [29,30]. In the endometrium, RA action is crucial for proper decidualization [31], whereas in the mammary gland, RA signaling plays a vital role in ductal morphogenesis [32]. The data reveal higher concentrations of RA in CGCs compared with these other ROL metabolizing tissues, indicating that CGCs are a major target for retinoid uptake and a significant source of RA production in the ovary. These studies demonstrated that CGCs are a primary source of follicle RA biosynthesis, and that higher mean RA levels in the CGCs of IVF patients yielded the highest percentage of successfully fertilized oocytes [21]. These findings were supported by Read and Dyce [22], who showed that bovine cumulus–oocyte complexes exposed to RA during maturation caused a significant increase in their maturation, cleavage, and blastocyst rates.

## 3. RA Can Promote Oocyte Competency via Regulation of Cx43 in CGC

It has been suggested that RA may enhance the cytoplasmic maturation of oocytes by modulating the expression of certain genes in CGCs, namely cyclooxygenase-2, midkine, nitric oxide synthase, and/or gonadotropin receptors [13]. Oocyte competency, known to correlate with efficient gap junction intercellular communication (GJIC) among granulosa cells in the cumulus–oocyte complex, involves Cx43 as the main subunit of gap junction channels in human CGCs. As such, Cx43 plays a crucial role in regulating the oocyte’s micronutrient environment by facilitating the transfer of ions, metabolites, and small molecules up to 1 kDa [10,11]. While Cx43 appears to be the only connexin contributing to gap junctions between the CGCs of growing follicles [33,34], Cx37 is the main connexin expressed in oocytes that links them with the CGCs. Although the identity of the connexin in the granulosa cells that form the oocyte–granulosa channels is not known in humans, in mice, those gap junctions were found to be composed of Cx37 in the oocytes linked with Cx37 exclusively expressed at the trans-zonal projections of the CGCs [35,36]. Multiple studies have demonstrated that Cx43 is needed in CGCs in order for the normal progression of folliculogenesis to take place and that the increased expression of this connexin, but not other connexins, is linked to the improved developmental competence of the oocyte [11,37]. In addition, patients with higher Cx43-expressing CGCs were found to have more successful embryo transfer and implantation rates and were more likely to have a successful pregnancy outcome [11]. Ovarian Cx43 has been shown to be regulated by luteinizing hormone (LH); the exposure of pre-ovulatory follicles to LH deactivates gap junctions through the induced phosphorylation of Cx43 [38,39]. This activity hinders the intercellular transport of cAMP, a meiosis inhibitor, leading to the resumption of meiosis [40]. The importance of this activity was emphasized by studies showing that Cx43 knockout mice have follicles that are unable to proceed beyond the pre-antral follicular stage [41]. In addition to its expression level, the phosphorylation status of Cx43 plays a crucial role in processes influencing GJIC, including gap junction assembly and channel gating [42]. The overall effects of Cx43 phosphorylation on GJIC have been shown to be dependent on the cell type as well as the specific amino acid residues that are modified. For example, most studies have shown that the phosphorylation of Cx43 by PKC in human cardiac cells correlates with reduced GJIC [42,43,44,45]. In contrast, in guinea pig cardiomyocytes and transfected HeLa cells, there are reports of increased GJIC following PKC activation [46]. Interestingly, Cx43 within the same species and cell type can exhibit opposite effects depending on the residues involved. Thus, in human cardiac cells, the phosphorylation of Cx43 at serine (S) residues S364, S365, and S369 increased GJIC, while that at residues S262, S368, and S372 reduced GJIC [47]. In contrast to that seen in cardiac cells, the phosphorylation of Cx43 at S368 in folliculostellate cells resulted in increased GJIC [48]. Studies from our laboratory showed that RA can upregulate Cx43 activity and GJIC in human endometrial stromal cells (ESC) and CGCs through a non-genomic mechanism that involves the rapid dephosphorylation of Cx43 (Figure 2) [21,24]. In ESC, this action was shown to be mediated through the increased interaction of Cx43 with its primary phosphatase, protein phosphatase 2A (PP2A) [24] (Figure 3). A similar mechanism of action involving the reduced phosphorylation of the ERK- and Akt-mediated pathways via the actions of PP2A has been demonstrated for the RA reduction in interferon-gamma and nitric oxide production in certain animal cells [49,50]. This non-genomic mechanism is in contrast to the canonical/genomic pathway of action by which RA is chaperoned into the cell nucleus by a distinct set of RA-binding proteins (CrabpI, CrabpII, Fabp5) to activate gene transcription via its binding with RXR-RAR heterodimers (Figure 3). The RA-induced dephosphorylation of Cx43 in ESC was shown to be at S262 [24], a residue of Cx43 whose phosphorylation is implicated in the GJIC-inhibitory action of certain growth factors (e.g., epidermal growth factor) and other PKC activators [51]. This action appears to be distinctive of ESC and CGC since the RA-induced dephosphorylation of Cx43 has not yet been detected in other cell types. The determination that RA rapidly increases GJIC in CGCs through this action on Cx43 provides a mechanism by which follicles and CGCs containing higher levels of RA can increase the probability of generating high-quality grade 1 embryos. This hypothesis has been supported by studies showing that cumulus–oocyte complexes exposed to RA in culture had significantly higher Cx43 expression correlating with increased gap junction coupling, and an increase in maturation, cleavage, and blastocyst rates [22].

## 4. A Role for RA in Endometriosis-Associated Infertility via Action on Cx43

Endometriosis impacts over 8 million women in North America alone, giving rise to various symptoms, with chronic pelvic pain and infertility being the most common [52,53]. In IVF procedures, women with endometriosis ovulate fewer oocytes than healthy counterparts, and the oocytes ovulated are often compromised [54,55,56]. Consequently, the fertilization and embryo cleavage rates after IVF, in both stimulated and unstimulated cycles, are significantly lower in women with endometriosis compared to healthy controls. The fertilization and embryo cleavage rates of endometriosis oocytes remain impaired even when sperm from their partners are substituted with sperm from donors [57]. Moreover, the implantation rates of oocytes from donors with endometriosis are reduced in recipients without endometriosis [58]. It has been established that follicular fluid from women with endometriosis undergoing IVF has a notably lower mean concentration of RA than control participants [28]. In contrast, the ROL concentrations do not differ between the two groups, indicating that the impairment of RA biosynthesis in the follicle, rather than the uptake of the ROL precursor, is responsible for the reduced RA levels. This finding is consistent with earlier data showing that, in endometrial cells from endometriosis patients, the dysregulation of the retinoid metabolic pathway results in the reduced production of RA and plays a fundamental early role in the ability of the cells to implant and grow at ectopic sites [29]. In those studies, it was determined that a very early event in the ability of endometrial cells to form ectopic lesions is the reduced expression of cellular retinol-binding protein type 1 (RBP1; also known as Crbp1), a retinol chaperone protein that serves as the preferred substrate for retinol dehydrogenase enzymes in the rate-limiting step in RA biosynthesis [59] (Figure 3). Studies from our group have shown that the expression of RBP1 is important for maintaining a normal flux through the retinoid biosynthetic pathway and is reduced in ESC from endometriosis patients [29]. Consistent with the role of RA in regulating Cx43 expression, preliminary studies have suggested a direct correlation between reduced Cx43 expression in ESC from endometriosis patients and the diminished ability of the cells to synthesize RA from retinol (Figure 4). If the same relationship between RA production and Cx43 expression occurs in the CGCs from these patients, this correlation would provide a mechanistic pathway by which reduced RA levels contribute to the decreased fecundity that is associated with this disease.

## 5. Concluding Remarks

Our understanding of the role played by RA in the acquisition of oocyte competence may provide critical information for developing novel non-invasive approaches for selecting high-quality oocytes for vitrification and/or use in single embryo transfers. As such, we propose studying CGC RA production as a predictor of oocyte competence. Recent improvements in assessing RA activity now make it possible to evaluate the activity in individual follicles. In previous studies, we and others have successfully used an Aldefluor assay to quantify the relative levels of cellular RALDH activity as a surrogate marker of RA biosynthesis to determine the major RA-producing cells in heterogeneous cell populations [60,61]. The Aldefluor reagent is a fluorescent substrate for ALDH, which is converted by ALDH into a membrane-impermeable isoform that is retained inside the cells and can then be measured by flow cytometry to determine the relative RA-producing capacity of cell subsets. In our previous work to determine the major RA-producing cells in human decidua, the LC-MS/MS analysis of RA levels in the cell populations validated the correlation between ALDH activity and their retinoid-synthesizing capacity. As recent studies have indicated that RA synthesis in animal and human ovarian granulosa cells is also dominated by ALDH [62], we hypothesize that ALDH assays could be utilized for determining the relative CGC RA levels obtained from individual follicles. Although an Aldefluor assay was used in the previous work, we suggest using an alternative ALDH colorimetric assay due to the limited number of CGCs in single follicles, as well as methodological considerations that would be required to obtain single cell suspensions that are suitable for flow cytometry. To this end, the mean number of human CGCs that surround an oocyte has been shown to be ~14,000 cells [63], which is compatible with commercially available ALDH colorimetric assays (e.g., Sigma-Aldrich ALDH Assay Kit, St. Louis, MO, USA). These commercial kits allow for a simple, fast, and reliable method to quantify ALDH enzymatic activity that can be utilized for evaluating multiple CGC samples harvested from patient follicles during IVF procedures.

The goal of assessing CGC RA production as a predictor of oocyte competence is especially appealing for two reasons. First, we know that RA plays a critical role in the acquisition of oocyte competence [13,16,17,18,19,20,64], and thus may mark that competence. This enables the performance of individual oocytes to be evaluated and correlated with the CGC RA levels and relevant clinical outcomes. Second, RA is on the causal chain influencing oocyte competence and thus may also be considered in therapeutic applications. Studying a biomarker that is also on the causal chain for producing healthy oocytes is an appealing approach. Through its action on Cx43, RA levels affect the formation of tight gap junctions within the cumulus–oocyte complex that promote the transfer of essential materials into the oocyte and improve their potential to produce a healthy pregnancy. Thus, RA is more than just a biomarker; it may also be a driver of the acquisition of that competence, offering the possibility that it could also become a treatment as well. In such a case, co-treatment with RA during folliculogenesis could improve patient outcomes. Although RA and its derivatives (e.g., isotretinoin, acitretin) have been shown to be efficacious and safe in a variety of clinical applications [65,66,67], they are highly teratogenic and they should only be used under the strict monitoring of pregnancy prevention. However, the natural RA compound (all-trans retinoic acid) has a short half-life of only 1–2 h [68] and could therefore be used during ovarian stimulation (8–14 days), withheld upon the hormonal trigger, and be gone by oocyte retrieval (~36 h after the trigger shot), thereby eliminating any concern about teratogenesis. The treatment of acute promyelocytic leukemia (APL) with orally administered all-trans RA provides an example of its clinical use in human subjects [69]. This includes the successful treatment of APL in pregnancy with all-trans RA [70]. Clearly, the safety profile of this potential therapeutic application to improve IVF patient outcomes will need to be assiduously vetted in animal studies before clinical trials in human pregnancy can be considered. We look forward to these future studies with enthusiasm.

## Figures and Tables

**Figure 1 ijms-25-01709-f001:**
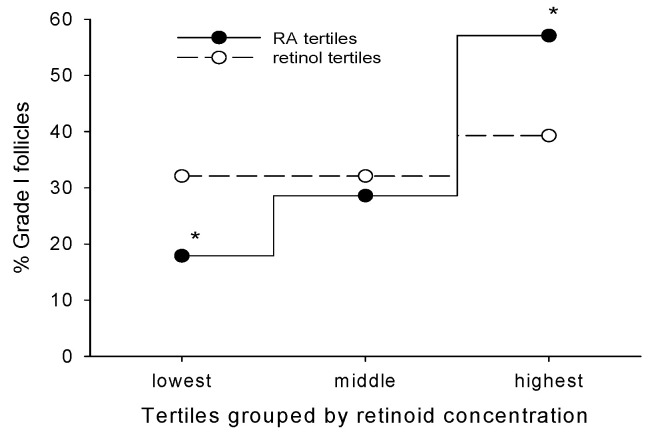
Percentage of grade I embryos derived from “mature egg follicles” (n = 84) stratified by tertiles according to RA or ROL concentrations. Percentage of grade I follicles = # of grade I follicles/28 (number of mature egg follicles in each tertile). * Significant difference between values at *p* < 0.005. RA concentrations in pmol/mL for each tertile (mean ± s.d.): lowest, 2.3 ± 1.2; middle, 4.8 ± 0.4; highest, 8.8 ± 3. Reproduced from Figure 3 from Pauli et al. [28].

**Figure 2 ijms-25-01709-f002:**
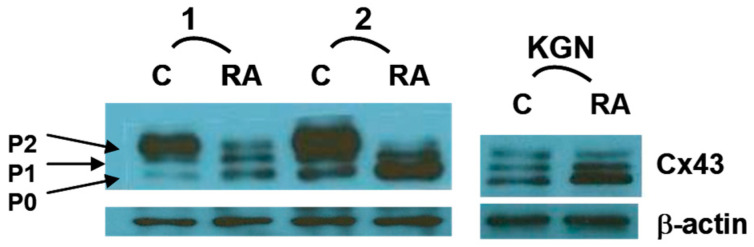
Effects of all-trans retinoic acid (RA) on Cx43 phosphorylation in cumulus granulosa cells. Total cellular protein was isolated from CGCs and the human granulosa cell line KGN and assayed for Cx43 expression by Western blotting. Representative experiments on primary CGCs from two patients and KGN cells treated with 10 µM of RA or vehicle control (C) as indicated for 48 h, with show changes in the band intensity and distribution of the non-phosphorylated (P0) and phosphorylated (P1 and P2) species of Cx43. The non-phosphorylated P0 band is the most biologically active form of Cx43 in these cells. Reproduced from. Figure 3A inset from Best et al. [21].

**Figure 3 ijms-25-01709-f003:**
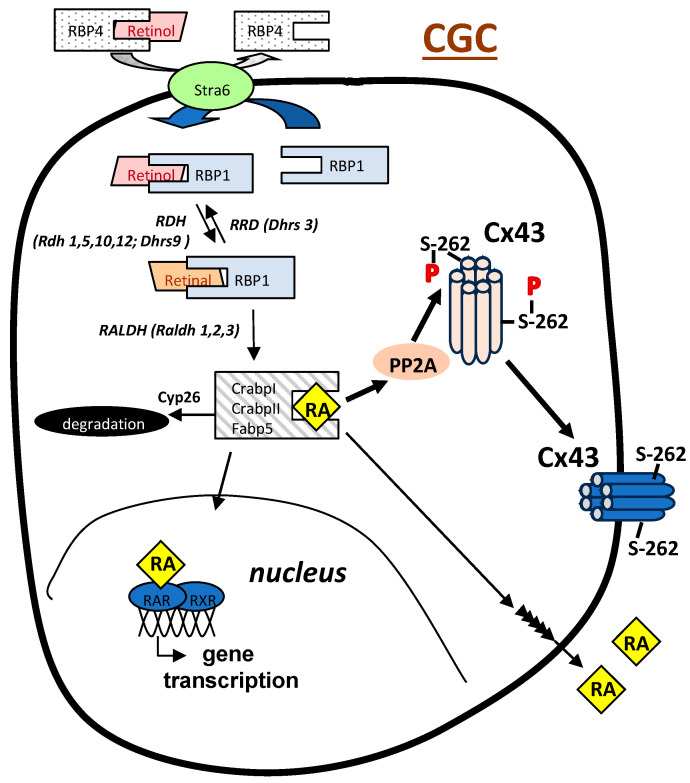
Schematic of retinoid metabolism and RA regulation of Cx43. The uptake of retinol (ROL) from circulating RBP4-bound retinol is regulated by the membrane receptor for vitamin A, Stra6. Intracellular chaperone cellular retinol-binding protein, type 1 (RBP1) physically interacts with Stra6 to pick up retinol and then delivers it to retinol dehydrogenase enzymes (RDH), which reversibly catalyze the conversion of retinol to retinal. RBP1 then chaperones retinal to retinal dehydrogenases (RALDH), which irreversibly convert retinal to RA. For the canonical/genomic regulation of various target genes, RA is then chaperoned into the cell nucleus by specific RA-binding proteins (CrabpI, CrabpII, Fabp5) to activate gene transcription. To regulate CGC gap junction activity, RA acts through a non-genomic mechanism to dephosphorylate Cx43 at the serine 262 (S-262) residue. This action is mediated through the increased interaction of Cx43 with its primary phosphatase, protein phosphatase 2A (PP2A). Once formed, RA can be (1) utilized in the RA-producing cell; (2) transported to neighboring cells (e.g., oocyte) to initiate RA-mediated signaling; and/or (3) degraded. RBP1 has been shown to be reduced in endometrial stromal cells from endometriosis patients leading to reduced RA production and Cx43 expression.

**Figure 4 ijms-25-01709-f004:**
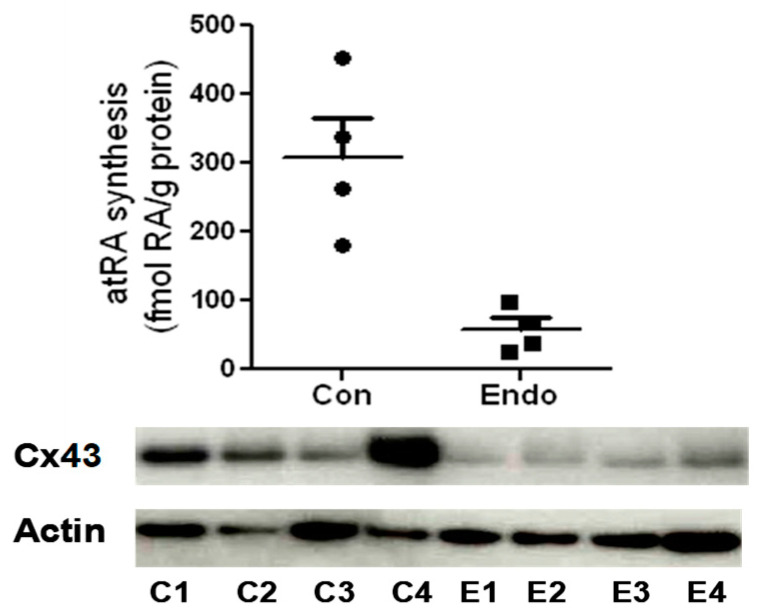
Synthesis of RA (top) and Western blotting of Cx43 (bottom) in endometrial stromal cells from the eutopic endometrium (proliferative phase) of four control (C) and four endometriosis patients (E). RA levels were quantified in cells by liquid chromatography–tandem mass spectrometry (LC-MS/MS) following treatment with 2 µM of retinol for 18 h [21,28]. In the absence of retinol, RA was not detected in all cases (unpublished data).

## Data Availability

Data presented can be found in referenced publications.

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
