# Peer review of "Retinoic Acid Action in Cumulus Cells: Implications for Oocyte Development and In Vitro Fertilization"

_ijms, 2024, doi:10.3390/ijms25031709_

Round 1
Reviewer 1 Report
Comments and Suggestions for Authors
This study aimed to provide evidence, based on the literature, that retinoic acid (RA) is associated with improvements in oocyte maturation due to its ability to regulate the activity of connexin 43 (Cx43) and the consequence of that.
I consider the manuscript to be well-written and able to provide important information in the field since oocyte maturation is the main key to IVF success. RA was already associated with improvements in oocyte maturation and quality in cattle, goats, pigs, camel, and mice, showing a positive impact on embryo production and quality in many of them. The interesting aspect of this manuscript is mainly the suggestion of new pharmacological approaches to optimize IVF outcomes and the possibility of selecting better oocytes in a non-invasive way based on the findings.
However, I believe the authors should be clearer in the abstract and introduction regarding the aim of the study. This is a hypothetical study based on a literature search review focused on establishing the association of RA with Cx43 to explain its positive effects and propose a noninvasive oocyte selection method. As a result, the authors also showed the possibility of pharmacological approaches using RA to improve IVF. The insights obtained from this study are good, but the main aim and the tools used to reach the aims need to be better described in the text.
Did the authors consider performing a systematic review to evaluate all the data obtained regarding RA and its impact on maturation to strengthen the hypothesis?
Reviewer 2 Report
Comments and Suggestions for Authors
The authors evaluated the possible effects of the presence of retinoic acid on oocyte development, cumulus cell expansion, and fertilization in vitro. The subject of this study is suitable for the “International Journal of Molecular Sciences” journal. Based on the results of previous studies, the authors claim that retinoic acid has significant positive effects on oocyte maturation and function by increasing connexin 43 activities, which is associated with the developmental competence of the oocyte. This hypothesis article is well prepared, and all processes and pathways that affect retinoic acid on oocyte maturation and fertility are clearly explained. Therefore, the manuscript can be accepted for publication in the present form.
The subject is original because, in addition to morphological evaluation, non-invasive molecular markers are needed to determine quality embryos, especially in vitro embryo production, which is one of the assisted reproductive techniques in humans. Thus, the success rate in in vitro embryo production and the birth of healthy offspring after transfer can be further improved.
The authors provide new information that retinoic acid can improve oocyte developmental competence and improve high-quality embryo ratio after fertilization in in vitro embryo production system.
The conclusions are consistent with the evidence and arguments.
The references are appropriate.
Figures are presented clearly and understandably.
Reviewer 3 Report
Comments and Suggestions for Authors
1. The hypothesis is stated, but it could be clearer. How can this research impact human IVF outcomes? Can this be added to the media and oocyte maturation media as an exogenous component? If yes, how these need to be prepared. Does it need approval for using human application? How much is shelf life of RA. What is the solvent of RA and at which timepoint it hould be used?
2. The manuscript provides a good overview of the challenges with different angles, but a stronger literature review of RA in other cells and tissue types would be even good. What other types of cell that RA has significant functions and what was the outcome? Why there is not fully agreement regarding the concentration and mechanism of RA?
3. While the study establishes a correlation between RA levels and oocyte quality, are there mechanistic insights explaining how RA can impact embryo directly or indirectly as well?
4. Is there potential for the assessment of RA levels to become a routine procedure in IVF clinics, nd what challenges might be associated with implementing such a measure?
5. Are there advantages or limitations in using RA as a marker compared to other potential indicators?
